# Prevalence of undernutrition and its associated factors among orphans aged 6–59 months in Nekemte town, Ethiopia

Werku Etafa[1]*, Fayera Abdana[2], Emebet Bobo[1], Wandimu Muche Mekonen[1], Dawit Tesfaye Daka[1], Meseret Belete Fite[3], Tesfaye Shibiru[4], Asefa Negeri[5], Gutu Leta[6], Dereje Temesgen[1]

1 Department of Pediatrics and Child Health Nursing, Institute of Health Sciences, Wollega University, Nekemte, Ethiopia, 2 Gidda Ayana General Hospital, East Wollega Zone, Oromia Regional State, Oromia, Ethiopia, 3 Department of Public Health, Institute of Health Sciences, Wollega University, Nekemte, Ethiopia, 4 Department of Pediatric and Child Health, School of Medicine, Institute of Health Sciences, Wollega University, Nekemte, Ethiopia, 5 Department of Emergency and Critical Care, School of Medicine, Institute of Health Sciences, Wollega University, Nekemte, Ethiopia, 6 Department of Anesthesia, School of Medicine, Institute of Health Sciences, Wollega University, Nekemte, Ethiopia

* witafay@gmail.com

## Abstract

### Background

Undernutrition is a leading cause of childhood mortality in low-income countries. Orphaned children are particularly vulnerable due to the absence of parental care and support, as well as factors such as infection and poor feeding practices. Undernutrition is often associated with developmental delays and recurrent infections. This study aimed to determine the prevalence of undernutrition and its associated factors among orphaned children aged 6–59 months in Nekemte town, Ethiopia.

### Methods

This study employed a cross-sectional, community-based study collected data through interviewer-administered questionnaires and anthropometric measurements. A total of 373 orphans paired with caregivers were selected using a simple random sampling technique from first June to July 30, 2023. Data were coded and entered into Epi Data V.4.6, and SPSS version 25 was used for analysis. Anthropometric indices were measured using the WHO ENA Software. Both binary and multivariate logistic regression analyses were conducted with a 95% confidence interval to identify factors associated with undernutrition, considering a significance level of $p < 0.05$.

### Results

The prevalence of stunting, wasting, and underweight in the studied population was 37.3% (95% CI: 32.4–41.8), 28.7% (95% CI: 23.6–33.2), and 24.4% (95% CI: 20.1–28.9),

**Data availability statement:** All relevant data are within the paper and its Supporting Information files.

**Funding:** The author(s) received no specific funding for this work.

**Competing interests:** The authors have declared that no competing interests exist.

**Abbreviations:** DDS, dietary diversity score; ENA, emergency nutritional assessment; FANTA, food and nutrition technical assistance; HFIAS, household food insecurity access scale; MUAC, mid upper arm circumference.

respectively. Factors associated with stunting included caregivers who obtained the child not through a legal process (AOR = 1.74, 95% CI: 1.06–2.86) and children who did not receive vitamin A in the last six months (AOR = 0.59, 95%CI: 0.371–0.94). For wasting, significant associations were found with orphans aged 6–23 months (AOR = 2.76, 95%CI: 1.27–6.02), those receiving treatment (AOR = 0.45, 95% CI: 0.26–0.76), incomplete vaccination status (AOR = 3.53, 95% CI: 1.65–7.04), and caregivers lacking information about nutrition for children under five (AOR = 1.82, 95%CI: 1.08–3.06). Additionally, orphans who began consuming additional food before six months of age (AOR = 2.76, 95%CI: 1.27–6.02) and those with caregivers who were government employees (AOR = 1.89, 95% CI: 1.07–3.34) were more likely to be underweight.

## Conclusions

The prevalence of undernutrition among orphaned children is high in Nekemte town. It is crucial to improve the knowledge and communication skills of healthcare workers and caregivers regarding infant and young child feeding practices and vitamin A supplementation for children under five years of age. Additionally, improving the supervision of orphans by legal bodies is essential for better health outcomes.

## Introduction

An orphan is a child whose mother, father, or both have passed away due to various conditions [1]. More than 150 million orphaned children are estimated worldwide, with approximately 17.8 million are double orphans [2]. In Ethiopia, more than five million children are orphaned, with 77,000 of them living in households where the child is the head. According to the Ethiopian Demographic and Health Survey (EDHS) 2016, 72% of children in the population live with both parents, while 14% live with their mothers alone, 11% do not have any biological parents, and 3% live with their fathers alone [3].

Undernutrition is defined as insufficient consumption of energy and nutrients to meet an individual's demands for optimal health [4]. It is the primary cause of death in children, accounting for 3.1 million (45%) fatalities worldwide [5]. Approximately 12% of Ethiopia's overall child population are orphans, and some of those under five years of age have stunted growth [3]. According to Ethiopia's malnutrition trend, the prevalence of stunting has reduced from 58% to 38%, and that of underweight has fallen from 41% to 24% over the last 15 years (2000–2015). However, the prevalence of wasting has marginally declined during the same period, from 12% to 10% [4,5].

The prevalence of undernutrition among orphaned children in Ethiopia varies depending on the setting [6,7]. According to four studies [8–11], at least one in every three orphan children between the ages of six and 59 months is stunted. In contrast, the lowest rate was recorded in Gambella (12.2%) [12]. However, the rates of wasted orphan children of the same age (6–59 months) are not significantly different from

each other (4.4% to 11.1) [8–11] and peak at a worrying 37.8% in Gambella [12]. Additionally, three studies revealed that at least one child in five is underweight [8,10,12]. Notably, investigations conducted in orphanages in Ethiopia and Addis Ababa yielded comparable results, with underweight rates of approximately 12% [9,11]. These data highlight the critical need for targeted interventions to address the dietary issues experienced by orphaned children throughout the country.

Despite being designated as a priority development issue in the context of the 2030 Agenda for the Sustainable Development Goals (SDGs) Target Framework, malnutrition remains a public health issue and one of the leading causes of infant morbidity and mortality in both developed and developing countries [13]. Stunting affects 150.8 million children worldwide, and while there has been some progress in lowering this condition, 50.5 million children are still wasted [14]. Undernutrition has a substantial influence on young orphans: it delays their physical growth, lowers their intellectual quotient, causes behavioral problems, causes a deficiency in social skills, and increases their susceptibility to contracting diseases [15,16]

Children in under five years of age in orphanages face nutritional challenges because the majority have not been nursed or are solely breastfed, and they are not offered a balanced diet for proper growth and development [17]. This renders them susceptible to infections because they have not received all the nutrients necessary for proper growth. [18]. Undernutrition enhances the frequency and severity of diseases, contributing to delayed recovery [19]. Furthermore, the interplay between malnutrition and disease can create a potentially fatal cycle of worsening illness and decreasing nutritional status. Fluctuations in nutritional status have a wide range of consequences on the body, including changes in organ size, hormones and cytokines, immune cell populations, and function [20].

The risk factors for undernutrition among orphan children include several interconnected issues related to feeding practices, such as the late initiation of complementary feeding and a meal frequency of less than three times per day. Additionally, household food insecurity, family size, and caregivers' age and educational status play significant roles. Recent illnesses, particularly diarrhea, poor knowledge and attitudes of caregivers, and a lack of vitamin A supplementation, further exacerbate the situation. Orphaned children under the age of two, double orphans, and those who have not received vaccinations are particularly vulnerable to undernutrition [3,8–12,21,22].

Different strategies have been implemented to resolve undernutrition in orphan children, but it remains a significant global public health issue despite existing interventions to address it [23,24]. Eliminating undernutrition in Ethiopia would avoid losses 8–11% annually from the gross national product [25]. This study was designed to assess undernutrition and related factors among orphan children aged 6–59 months, which may be utilized as a reference in defining priorities and designing effective nutritional programs by any organization or agency focusing on this topic. To our knowledge, no study has revealed the undernutrition of community-based under-five orphan children in Nekemte town. As a result, this research would assist in fulfilling a gap by providing information to the concerned bodies about orphanage nutrition.

## Methods

### Study setting, period and populations

A community-based cross sectional study was conducted in Nekemte town from first June to July 30, 2023. Nekemte is situated in the East Wollega Zone, 331 km from Addis Ababa, the capital city of Ethiopia. The number of under-five orphan children (U5) in Nekemte town Aged 6–59 months are six hundred sixty-three (663); 295 (males), and 368 females. The target population for this research is all orphaned children aged 6–59 months paired with adoptive parents residing in Nekemte.

### Eligibility

The study excluded orphan children who were seriously ill, had physical deformities not suitable for anthropometric measurements, or whose caregivers had communication difficulties due to impairments or disablities.

## Sample size determination and sampling procedure

We estimated the sample size using the single population proportion based on a 5% margin of error, a 95% confidence interval, and the prevalence of stunting from the study conducted in Addis Ababa (34.8%) among orphans aged 6–59-months [9]. Based on these assumptions, we obtained 349 samples for analysis. By adding 10% for non-response rates, the final sample size was 384.

The study participants were selected using simple random sampling. We employed a lottery method based on registration books to generate a sampling frame. The registry was obtained from the Town Youth and Women's office. When two or more orphans were discovered in the same household, the younger child was chosen for the study.

## Study variables

The dependent variable is the nutritional status of orphan children aged 6–59 months. The independent variables include socio-demographic characteristics, environmental and behavioral factors, feeding practices, health-related factors, and access to social and legal services.

## Operational definitions

**Orphan:** is a child who has lost one or both parents through death [26].
**Undernutrition**: undernutrition results from inadequate intake of micro and macronutrients and is classified as stunting, wasting, underweight and micronutrient deficiencies [27,28].
**Stunting:** Is chronic form of undernutrition, expressed as height/length-for-age <-2SD from the median height/length in the age of the reference group [28,29].
**Wasting:** Is an acute form of undernutrition, expressed as weight-for-height or weight-for-length < −2SD from the median body weight for the height or length in the reference group [28,29].
**Underweight:** a composite form of undernutrition including stunting and wasting, and expressed as weight-for-age < −2SD from the median body weight for the age of the reference group [28,30]
**Presence of illness:** was assessed by asking children/caregivers of children whether they had symptoms of cough, diarrhea, fever and vomiting within the previous two weeks before the survey [31]
**Food security:** A situation that exists when all people, at all times, have physical, social and economic access to sufficient, safe and nutritious food that meets their dietary needs and food preferences for an active and healthy life [32,33].
**Food insecurity:** A person is food insecure when they lack regular access to enough safe and nutritious food for normal growth and development and an active and healthy life. This may be due to unavailability of food and/or lack of resources to obtain food [32,33].
**Individual dietary diversity score:** The dietary group intake frequency score (FGFS) was determined by assigning a score of 0 if not ingested within the preceding 24 hours and 1 if consumed. Children between the ages of six months and five years had a good DDS score of ≥4 out of 9 and a poor DDS score of <4 [34,35].

## Data collection and quality control

A data collection tool was developed following a comprehensive review of relevant literature and previous studies, organized according to specific objectives [30,36,37] (Supporting Information 1). A pre-tested, structured interviewer-administered questionnaire was utilized, originally written in English and subsequently translated into Afaan Oromoo. This questionnaire gathered information from the guardians of children regarding demographic factors, feeding practices, food availability, family size, vaccination status, periodic food intake, exposure to diarrhea, and instances of acute febrile illness.

Household food security was evaluated using the Household Food Insecurity Access Scale, a validated tool developed by the Food and Nutrition Technical Assistance. This tool consists of nine occurrence questions and an additional nine frequency questions, which together assess the severity of household food insecurity over the past four weeks [38].

## Measurements

Anthropometric measurements, including height, weight, and mid-upper arm circumference (MUAC), were obtained using standardized measurement tools and established operational protocols. Children under two years of age were weighed on a spring scale, without shoes and dressed in light clothing, to the nearest 0.1 kg. For children over two years, weight was measured using a beam balance, with participants barefoot and also dressed in light clothing. To ensure precision, the scales were calibrated immediately prior to and during each session using a standardized calibration weight of 5 kg.

Before measurement, the child's shoes, hair clips, and braids are removed. The child is positioned with feet together, flat on the ground, with heels touching the back plate of the measuring tool. The legs should be straight, the buttocks against the backboard, the scapula in contact with the backboard, and the arms relaxed at the sides. The mid-upper arm circumference (MUAC) is measured using a colour-coded standard MUAC tape by identifying the midpoint between the shoulder and elbow. This measurement is taken at the midpoint and recorded to the nearest 0.1 cm. The data were then entered and processed using the ENA Smart version 2022 software to standardize the anthropometric measurements.

A pre-test was conducted on 5% of the sample size from the orphan population, which was excluded from the main study. The weighing scale was calibrated to ensure the scale points were at zero before taking any readings. Training was provided on interview methodologies, sampling processes, inclusion and exclusion criteria, sources of information, error reduction techniques, and anthropometric measurements. The measurement scale used is a valid and standardized tool adapted from the Food and Nutrition Technical Assistance (FANTA) instruments. The data collection team comprised one supervisor (a public health officer), two BSc nurses, and twelve health extension workers, all familiar with the designated orphanages. Training was provided covering the research aims, questionnaire content, privacy issues, measurement techniques, and anthropometric protocols. All measurements were conducted following standard procedures, with explanations provided to the mothers or caretakers of the children.

## Data processing and analysis

The data underwent verification for accuracy and completeness before being coded and imported into Epi Data V.4.6 software. It was subsequently exported to SPSS V.25 for analysis (Supporting Information 2). Nutritional indicators, including height/length-for-age Z-scores, weight-for-age Z-scores, and weight-for-height/length Z-scores, were calculated based on WHO 2006 child growth standards. The WHO Emergency Nutritional Assessment (ENA) Software generated the Z-scores, categorizing children as stunted (HAZ < −2 SD), underweight (WAZ < −2 SD), or wasted (WHZ - < 2 SD).

Model fit was assessed using the Hosmer-Lemeshow test, yielding values of 0.163, 0.496, and 0.487 for stunting, wasting, and underweight, respectively, indicating a good fit (all > 0.05). Multicollinearity was checked, with tolerance values above 0.2 and VIF below 10, confirming no issues. Summary tables were created for variable presentation. Bivariate logistic regression analyzed the relationships between predictors and outcomes (stunting, wasting, and underweight). Variables with a significance level of < 0.25 in bivariate analysis were eligible for multivariate logistic regression to control for confounders, retaining those with a p-value < 0.05 in the final analysis.

## Ethical considerations

Ethical clearance was obtained from the Wollega University Research Ethics Committee, and a permission letter was issued to all study areas within town administration. Participants were informed about the study's objectives, and their involvement was entirely voluntary. Each participant provided a written and verbal informed consent based on their

preference. To ensure confidentiality, respondents' and orphans' names, orphans' age, address, and date of birth are removed from the dataset and replaced with coded numbers (Supporting Information 2).

## Results

### Sociodemographic characteristics of orphans paired with caregivers

In this study, a total of 373 participants were interviewed, resulting in a response rate of 97.1%. Among them, 187 (50.1%) were female, and 239 (64.1%) were over 23 months old. The mean and standard deviation (Mean±SD) of age, weight and height of the orphans was 33.78±17.8 SD months, 12.99±3.25 kg, and 88.22±14.83 cm, respectively. Regarding guardians, 157 (42.1%) were aged between 30 and 40 years, and the majority of orphans, 248 (66.5%), had lost their fathers. Educationally, 47 (12.6%) of the respondents were illiterate, while 142 (38.1%) had completed basic education and 94 (25.1%) had finished high school. Additionally, 284 (76.1%) of the households had two or fewer children under five years old, whereas 89 (23.9%) had three or more (Table 1).

### Health status, environmental, behavioral, social and legal services-related characteristics

In this study, only 227 (60.9%) of orphan children were fully vaccinated. In the two weeks prior to data collection, 130 (34.9%) had experienced illnesses such as diarrhea, fever, and cough. Regarding nutrition, 166 (44.5%) had consumed four or more food groups. Most caregivers (79.1%) practiced handwashing with soap before feeding their children, and 352 (94.4%) did so after using the toilet. Approximately 62.5% of caregivers and orphan children accessed drinking water from piped sources. However, only 57 (15.3%) of the orphan children received social support. Notably, none of the children had legal follow-up (373; 100%) (Table 2).

### Dietary intake and individual dietary diversity score

The study found that the majority (65.1%) of participants indicated that the first food caregivers provided to orphan children was milk. Additionally, more than half (55.5%) of the orphan children had a low dietary diversity score, consuming fewer than four different food items. Most of the children (76.4%) had been breastfed, with over half (56.3%) exclusively breastfed (Table 3).

### Households food security

In terms of household food insecurity, over half of the household members (56.6%) reported consuming fewer meals each day. Additionally, about half (50.9%) of the caregivers have expressed their concern that they would not have enough food. Furthermore, nearly half of the household members (53.6%) indicated that they ate only a limited variety of meals daily due to resource shortages (Table 4).

### Prevalence of stunting, wasting, and underweight

In the current study, the prevalence of stunting among orphaned children was found to be 139 (37.3%, 95% CI: 32.4–41.8%). Additionally, 28.7% (95% CI: 23.6%–33.2%) of the children were classified as wasted, and 24.4% (95% CI: 20.1%–28.9%) were underweight (Fig 1).

### Factors associated with stunting, underweight and wasting among orphans aged 6–59 months

A multivariable logistic regression identified factors related to stunting, underweight, and wasting. Stunting in orphaned children is strongly and positively connected with orphans who have not taken vitamin A supplements in the previous six months and orphans adopted from individuals. The study found that orphans who did not receive vitamin A supplements had a 41% higher chance of developing stunting than those who did (AOR: 0.59, 95% CI: 0.371–0.94). Furthermore,

**Table 1. Socio-Demographic Characteristics of Orphans and Their Caregivers in Nekemte town, Ethiopia, 2023.**

| Variable | Category | Frequency (n) | Percentage (%) |
|---|---|---|---|
| Sex of the child | Male | 186 | 49.9 |
| | Female | 187 | 50.1 |
| Age of children (Mean±SD) (33.78±17.8) (months) | 6-23 | 134 | 35.9 |
| | 24-59 | 239 | 64.1 |
| Occupation of the caretakers | Housewife | 204 | 54.7 |
| | Government employ | 115 | 30.8 |
| | Merchant | 54 | 14.5 |
| Whom did the child lose | Father | 248 | 66.5 |
| | Mother | 52 | 13.9 |
| | Both father and mother | 73 | 19.6 |
| Age of respondent (years) | ≤ 30 | 152 | 40.8 |
| | 31-40 | 157 | 42.1 |
| | 41-50 | 64 | 17.2 |
| Where did the child stay during the day? | Guardian House | 193 | 51.7 |
| | School | 86 | 23.1 |
| | Others(neighborhood, relatives, workplace) | 94 | 25.2 |
| From whom did you take the child | Government body | 129 | 34.6 |
| | NGO | 78 | 20.9 |
| | Individual | 166 | 44.5 |
| Age of the child at the time you received | 6-11 month | 342 | 91.7 |
| | ≥12 month | 31 | 8.3 |
| Educational status | Not attended school | 47 | 12.6 |
| | Grade 1–8 | 142 | 38.1 |
| | Grade 9–12 | 94 | 25.2 |
| | College and above | 90 | 24.1 |
| Perception of caregivers about their monthly income | Sufficient | 33 | 8.8 |
| | Medium | 102 | 27.3 |
| | Insufficient | 238 | 63.8 |
| Relationship of the child with the caregiver | Nuclear family | 304 | 81.5 |
| | Neighborhood | 69 | 18.5 |
| Mean±SD weight of orphans in kilogram | 12.99±3.24 | | |
| Mean±SD Height/length in centimeter | 88.21±14.82 | | |

orphans adopted from individuals had nearly double the risk of stunting as those adopted from governing bodies (AOR: 1.74, 95% CI: 1.06–2.86).

In terms of wasting, the study found that orphaned children aged 6–23 months were 71% more likely to have this condition than those aged 24–59 months (AOR=0.29; 95% CI: 0.14–0.60). Furthermore, children who had not completed their vaccines were 3.53 times more likely to experience wasting than their fully vaccinated counterparts (AOR=3.53; 95% CI: 1.65–7.04). Orphans cared for by caregivers who did not have up-to-date nutrition information for children under five were 1.82 times more likely to waste than those whose caregivers were aware (AOR=1.82; 95% CI: 1.08–3.06). In contrast, children getting treatment for acute or chronic diseases were 55% less likely to be wasted (AOR=0.45; 95% CI: 0.26–0.76).

Children who began supplementary feeding immediately after birth were 2.76 times more likely to be underweight compared to those who started at six months or later (AOR 2.76, 95% CI: 1.27–6.017). Moreover, children whose caregivers

**Table 2. Health status, environmental, behavioral, social and legal services-related characteristics related characteristics of orphan children in Nekemte town, Ethiopia, 2023.**

| Variable | Category | Frequency (n) | Percentage (%) |
|---|---|---|---|
| Vaccination statues | Fully vaccinated | 227 | 60.9 |
| | Not completed | 55 | 14.7 |
| | up-to-date | 91 | 24.4 |
| The child suffered from illnesses in the previous two weeks | Yes | 130 | 34.9 |
| | No | 243 | 65.1 |
| The child treated for acute or chronic illnesses in last two weeks | Yes | 166 | 44.5 |
| | No | 207 | 55.5 |
| The child tested for malnutrition in the last 2 weeks. | Yes | 95 | 25.5 |
| | No | 278 | 74.5 |
| Do you have a vaccination card | Yes | 345 | 92.5 |
| | No | 28 | 7.5 |
| What vaccine has the child taken | BCG | 116 | 31.1 |
| | Polio | 65 | 17.4 |
| | Measles | 192 | 51.5 |
| Has the youngster received Vitamin A (in the last 6 months) | Yes | 246 | 66 |
| | No | 127 | 34 |
| Wash hand using | Only water | 124 | 33.2 |
| | Water with soap | 249 | 66.8 |
| Source of drinking water | Pipe | 233 | 62.5 |
| | Protected spring | 140 | 37.5 |
| Water storage method | Jerican | 323 | 86.6 |
| | Bucket | 50 | 13.4 |
| Wash hand by water and soap before feeding | Yes | 295 | 7.1 |
| | No | 78 | 20.9 |
| Frequently wash hand with water and soap after the toilet | Yes | 352 | 94.4 |
| | No | 21 | 5.6 |
| The child has social support | Yes | 57 | 15.6 |
| | No | 316 | 84.4 |
| Is there any support given to the orphan | Yes | 60 | 16.1 |
| | No | 313 | 83.9 |
| Does the child have legal follow-up | Yes | 0 | 0 |
| | No | 373 | 100 |

were government employees were 1.89 times more likely to be underweight than those cared for by housewives (AOR 1.89, 95% CI: 1.07–3.34) (Table 5).

## Discussion

The current study investigated the prevalence of undernutrition and its associated factors in orphaned children aged 6–59 months in Nekemte Town, Ethiopia. The prevalence rates of stunting, wasting and underweight were 37.3%, 28.7%, and 24.4%, respectively. Stunting was significantly associated with children who did not receive vitamin A supplementation and the organization that placed the child with adoptive parents. Wasting is linked to child's age, vaccination status, and treatment for acute or chronic diseases. Additionally, the age at which children begun supplementary feeding and the occupation of their caregivers were significant factors related to underweight status.

**Table 3. Dietary Intake and Individual Dietary Diversity Score Characteristics of Orphan Children in Nekemte town, Ethiopia, 2023.**

| Variable | Category | Frequency (n) | Percentage (%) |
|---|---|---|---|
| Did the child Breastfeed | Yes | 285 | 76.4 |
| | No | 88 | 23.6 |
| Did the child exclusively breastfeed | Yes | 210 | 56.3 |
| | No | 119 | 31.9 |
| | I don't know | 44 | 11.8 |
| The child began meals | Soon after birth. | 42 | 11.3 |
| | Within one to six months. | 134 | 35.9 |
| | Within six to twelve months | 197 | 52.8 |
| Initial food the child feed | Milk | 243 | 65.1 |
| | Adult foods | 44 | 11.8 |
| | Porridge | 71 | 19.1 |
| | Other specify | 15 | 4 |
| Feeding utensil | Hand | 91 | 24.4 |
| | Cups and spoons | 166 | 44.5 |
| | Bottle | 116 | 31.1 |
| Have information about U5C feeding from the health professional or trained individual | Yes | 226 | 60.6 |
| | No | 147 | 39.4 |
| Consumed any fat foods | Yes | 190 | 50.9 |
| | No | 183 | 49.1 |
| Consumed pumpkin, orange flesh sweet potatoes, potatoes, | Yes | 209 | 56 |
| | No | 164 | 44 |
| Consumed dark green leafy vegetables | Yes | 205 | 55 |
| | No | 168 | 45 |
| Consumed any fruits | Yes | 231 | 61.9 |
| | No | 142 | 38.1 |
| Consumed any flesh or organ meats | Yes | 63 | 16.9 |
| | No | 310 | 83.1 |
| Consumed milk and dairy products | Yes | 220 | 59 |
| | No | 131 | 35.1 |
| Consumed any foods containing oil, fat, or butter | Yes | 100 | 26.8 |
| | No | 273 | 73.2 |
| Consumed eggs | Yes | 267 | 71.6 |
| | No | 106 | 28.4 |
| Consumed any bean-based foods | Yes | 242 | 64.9 |
| | No | 131 | 35.1 |
| Consumed honey, sugar, or a sweet/soft drink | Yes | 148 | 39.7 |
| | No | 225 | 60.3 |
| Dietary diversity score | High intake (> 4) | 166 | 45.5 |
| | Low intake (≤ 4) | 207 | 55.5 |

In this study, the prevalence of stunting aligns closely with the findings from Kazakhstan (36.7%) [39], and Nigeria (34%) [40]. Furthermore, stunting rates in Addis Ababa, Ethiopia, were reported to be34.8% and 35.1% [9,11]. However, these figures are lower than those reported in Gondar and Addis Ababa, Ethiopia [8,10]. In contrast, the prevalence rates reported in a study conducted in Kenya (15.4%) [41], and Ethiopia in Gambella region (10%) [12] are lower than that of

**Table 4. Household food insecurity of orphan children living in Nekemte town, Ethiopia, 2023.**

| Variable | Category | Frequency (n) | Percentage (%) |
|---|---|---|---|
| Did you worry about your family not having enough food? | Yes | 190 | 50.9 |
| | No | 183 | 49.1 |
| If yes, how frequently? | Rarely | 45 | 23.7 |
| | Sometimes | 89 | 46.8 |
| | Often | 56 | 29.4 |
| Did you or any family member eat only a few different types of food on a daily basis owing to a shortage of resources? | Yes | 200 | 53.6 |
| | No | 173 | 46.4 |
| If yes, how frequently? | Often | 70 | 35 |
| | Rarely | 97 | 48.5 |
| | Sometimes | 33 | 16.5 |
| Did you or any of your household members eat fewer meals each day because there wasn't enough food? | Yes | 211 | 56.6 |
| | No | 162 | 43.4 |
| If yes, how frequently? | Often | 71 | 33.6 |
| | Rarely | 109 | 51.6 |
| | Sometimes | 31 | 14.8 |
| Did your home ever go without food because there weren't enough resources to get more? | Yes | 90 | 24.1 |
| | No | 283 | 75.9 |
| If yes, how frequently? | Rarely | 72 | 80 |
| | Sometimes | 18 | 20 |
| Did you or any family member go to bed hungry due to not having enough food? | Yes | 162 | 43.4 |
| | No | 211 | 56.6 |
| If yes, how frequently? | Rarely | 91 | 56.1 |
| | Sometimes | 71 | 43.9 |
| Did you or a family member go an entire day without feeding because there wasn't enough food? | Yes | 86 | 23.1 |
| | No | 287 | 76.9 |
| If yes, how frequently? | Rarely | 78 | 90.6 |
| | Sometimes | 8 | 9.4 |
| Did you or any other household member have a smaller meal than you felt was necessary due to a lack of food? | Yes | 123 | 33 |
| | No | 250 | 67 |
| If yes, how frequently? | Rarely | 8 | 6.5 |
| | Sometimes | 97 | 78.8 |
| | Often | 18 | 14.7 |
| Did your home ever go without food because there weren't sufficient resources to get more? | Yes | 4 | 0.8 |
| | No | 369 | 99.2 |
| If yes, how frequently? | Rarely | 4 | 100 |

the current study. Stunting is a chronic type of undernutrition resulting from prolonged exposure to food insecurity and chronic illnesses. Factors such as feeding skills and cultural practices, types of foods consumed, economic disparities, access to healthcare services, food availability, and the practices of organizations supporting orphans and vulnerable children significantly influence stunting rates, particularly in different geographic locations. Moreover, height-for-age or length-for-age measurements are affected by the chronicity of diseases and genetic factors. For example, genetically, the Gambella ethnic group in Ethiopia tends to be taller than other ethnic groups, which may explain the variation in stunting rates observed between Gambella (12.2%) and the current study (37.3%).

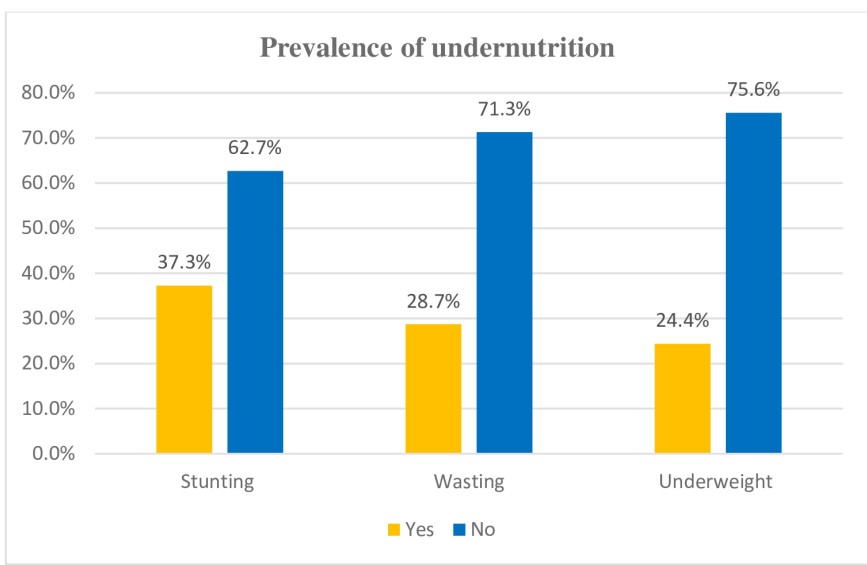

**Fig 1. Prevalence of undernutrition among orphan children aged 6-59months in Nekemte town, Ethiopia, 2023.**

Orphaned children who do not receive Vitamin A supplementation are 41% more likely to be stunted than those who are supplemented. This finding aligns with studies conducted in orphanage centers in Addis Ababa, Gondar City, and Gambella, found in Ethiopia [8,9,12]. This underscores the critical role of early Vitamin A supplementation in preventing stunting in this vulnerable population. To effectively combat undernutrition among orphaned children, it is essential to counsel mothers and other caregivers on the importance of early Vitamin A supplementation. Therefore, placing bodies, adoptive parents, and healthcare workers closely working with orphans need to be informed about the beneficial impacts of vitamin A supplementation on orphan growth and development. In addition, orphans adopted by individuals are at a higher risk of stunting than those adopted from governing or legal bodies. This finding is also supported by a systematic review [42]. This shows that orphan children have better opportunities if they are placed with adoptive parents legally. In this case, is due to the effect of either the legal bodies' follow-up or adoptive parents having a deep understanding of childcare.

The prevalence of wasting in this study is consistent with the findings reported in Gondar Town (27.8%) [8]. However, it is lower than the prevalence rates in northern India (62%) [43] and the Gambella region in Ethiopia (37.8%) [12]. Conversely, it is higher than that reported in Dilla town (11.1%) [44], and three studies identified the level of undernutrition: two studies based on orphanage centers and one study based on community based findings in Ethiopia [9,11,22]. The discrepancies in these prevalence rates may be attributed to variations in the age, physical condition, and physiological state of the children included in the studies. Wasting is an acute form of undernutrition primarily resulting from exposure to acute illnesses, which can vary over time and across locations, thereby influencing the prevalence rate of wasting among children aged 6–59 months. Additionally, factors such as environmental sanitation and hygiene play crucial roles in these discrepancies.

Receiving treatment for an acute or chronic illness reduced the probability of being wasted by more than half. This finding is supported by a study conducted among orphan children in orphanage centers in Addis Ababa, Ethiopia [9]. Wasting could be related to malnourishment and infection increases nutritional loss, decreases appetite and food intake, and eventually leads to undernutrition, which is one of the immediate causes of malnutrition [45]. Seeking quick treatment is one of the best ways to treat an illness. Orphan children aged 6–23 months were 71% more likely to be wasted than those whose age was24–59 months. This finding is supported by the study conducted in Ethiopia

**Table 5. Factors affecting stunting, underweight and wasting among orphan children aged 6-59 months in Nekemte town, Ethiopia, 2023.**

**Factors associated with stunting, underweight and wasting**

| Variables | Category | Stunting | | COR(95%CI) | AOR(95%CI) | p-value |
|---|---|---|---|---|---|---|
| | | Yes | No | | | |
| Took vitamin A | Yes | 101 | 145 | 1 | 1 | |
| | No | 38 | 89 | 0.61(0.388–0.95) | 0.59(0.371–0.94) | 0.03* |
| The child adopted from | Government body | 41 | 88 | 1 | 1 | |
| | NGO | 52 | 26 | 4.2(1.07-4.95) | 4.6(1.07-5.00) | 0.9 |
| | Individual | 72 | 94 | 1.64(1.016 −2.66) | 1.74(1.06–2.86) | 0.02* |
| During the day the child stays with | Guardian House | 78 | 115 | 1 | 1 | |
| | School | 19 | 67 | 0.41(0.233–0.75) | 0.41(0.21-0.78) | 0.07 |
| | Others | 42 | 52 | 1.19(0.724–1.95) | 1.15(0.68-1.93) | 0.25 |
| Duration of breastfed | Six months or less | 13 | 28 | 0.61(0.31–1.3) | 0.77(0.36-1.64) | 0.5 |
| | 6–11 months | 21 | 49 | 0.56(0.3- 1.0) | 0.68(0.36-1.21) | 0.22 |
| | 12–23 months | 18 | 43 | 0.54(0.3–1.02) | 0.64(0.33-1.21) | 0.17 |
| | 2 years and above | 87 | 114 | 1 | 1 | |
| Dietary diversity score | >4 | 54 | 112 | 1 | 1 | |
| | ≤4 | 85 | 122 | 1.44(0.943- 2.21) | 1.32(0.85-2.07) | 0.21 |

Factors affecting Wasting

| Variables | Category | Wasting | | COR(95%CI) | AOR(95%CI) | p-value No |
|---|---|---|---|---|---|---|
| | | Yes | No | | | |
| Child's age (months) | 6-23 | 17 | 117 | 0.24(0.136–0.42) | 0.29(0.14 −0.60) | 0.001* |
| | 24-59 | 90 | 149 | 1 | 1 | |
| Had information about child feeding | Yes | 52 | 174 | 1 | 1 | 0.02* |
| | No | 55 | 92 | 1.89(1.268-3.155) | 1.82(1.08-3.06) | |
| Vaccination status | Completed | 66 | 161 | 1 | 1 | |
| | Not completed | 24 | 31 | 1.89(1.031-3.458) | 3.5(1.65-7.04) | 0.001* |
| | Up to date | 17 | 74 | 0.56(0.308-1,021) | 1.16(0.56–2.38) | 0.05 |
| The child treated for acute or chronic disease | Yes | 57 | 109 | 1 | 1 | |
| | No | 50 | 157 | 0.60(0.388-0.957) | 0.45 (0.26 −0.76) | 0.003* |
| support given to the orphan children | Yes | 11 | 49 | 1 | | |
| | No | 96 | 217 | 1.97(0.98-3.95) | 1.05 (0.37 - 3.29) | 0.25 |
| The respondent's relationship with the child | Nuclear family | 93 | 211 | 1 | 1 | |
| | Neighborhood | 14 | 55 | 0.57(0.30–1.09) | 0.85(0.37-1,95) | 0.71 |
| Who did the child loss | Father | 81 | 167 | 1 | 1 | |
| | Mother | 13 | 39 | 0.69(0.35-1.36) | 1.6(0.66-3.94) | 0.28 |
| | Both father and mother | 13 | 60 | 0.45(0.23 −0.86) | 0.52(0.21-1.24) | 0.14 |

| Variables | Category | Underweight | | COR(95%CI) | AOR(95%CI) | p-value |
|---|---|---|---|---|---|---|
| | | Yes | No | | | |
| The age at which the child started additional food | Immediately after birth | 17 | 25 | 2.17(1.08–4.36) | 2.76(1.27–6.08) | 0.01* |
| | From first month to 6th month | 27 | 107 | 0.805(0.471.37) | 2.65(1.28-5.50) | 0.09 |
| | From 6th months to 12th month | 47 | 150 | 1 | 1 | |
| Occupation of the caretaker | Housewife | 44 | 160 | 1 | 1 | |
| | Government employ | 36 | 79 | 1.79(1.09 −2.77) | 1.89(1.07-3.34) | 0.027* |
| | Merchant | 11 | 43 | 0.93(0.44-1.95) | 1.48(0.86-2.55) | 0.15 |
| Age of the child in month | 6-23 | 11 | 123 | 0.17(0.09- 0.34) | 0.25(0.55-1.02) | 0.06 |
| | 24-59 | 80 | 159 | 1 | 1 | |

*(Continued)*

**Table 5.** (Continued)

| Factors associated with stunting, underweight and wasting | | | | | | |
|---|---|---|---|---|---|---|
| Number of U5Cin HH | <2 | 60 | 223 | 1 | 1 | |
| | ≥2 | 31 | 59 | 1.95(1.16 −3.28) | 1.51(0.84-2.73) | 0.16 |
| Did the child receive treatment for acute or chronic illnesses | Yes | 47 | 119 | 1 | 1 | |
| | No | 44 | 163 | 0.68(0.42-1.09) | 0.76(0.42-1.35) | 0.35 |
| Was the child tested for malnutrition? | Yes | 31 | 64 | 1 | 1 | |
| | No | 60 | 218 | 0.56(0.33-0.95) | 0.76(0.40-1.44) | 0.41 |

*: Statistically significant at p-value<0.05 in multivariable logistic regression

[46]. The transition from exclusive breastfeeding to complementary family food can expose a child to undernutrition as the child grows, and after the first six months, the breast milk becomes insufficient to meet nutritional needs [47]. This problem can occur if complementary foods not introduced as needed. According to the current study, orphaned children with incomplete vaccination are at a higher risk of wasting than those who are fully vaccinated. This finding is in line with a study conducted in Addis Ababa, Ethiopia [2]. Complete and current immunization is a public health prevention measure required to aid in the fight against infectious diseases in developing countries [48]. This could be because children lack or have a poor bodily defense mechanism, allowing them to be infected with a variety of infectious and vaccine-preventable diseases such as pneumonia, otitis media, measles, and diarrhea. As a result of not completing vaccination, children are more likely to be exposed to and contaminated by various infectious agents, which can lead to illness, and finally, undernutrition.

In this study, orphan children cared for by adopting parents who have information about child feeding are less likely to be wasted than their counterparts. This finding is similar to that of other studies [49,50]. This is the fact that individual families who have information on how to provide nutrition according to their age are quite different from their counterparts. Therefore, the quality of nutrition knowledge among health workers needs to be strengthened to provide high-quality and consistent nutrition information to caregivers of children with disabilities. Generally, caregivers who lack knowledge about feeding under-five orphan children are in danger, and instructions on orphan child feeding practices should be provided to caregivers.

The proportion of underweight children was similar to that reported in Addis Ababa (25.5%) [10]. Nonetheless, this is notably higher than previous findings from Ethiopia [8,9,11,12]. The socioeconomic status of the caregivers affects the nutritional status of orphaned children [44]. In this study, the majority 238 (63.8%) of adopting parents perceived their monthly income as insufficient. This implies that government bodies should not place orphan children with adoptive parents because of their interest; rather, a thorough economic investigation should be conducted prior to placement. Additionally, the research highlighted that the occupation of caregivers significantly impacts underweight rates; children with government worker caregivers were nearly two times more likely to be underweight than those cared for by house-wives. This may be attributed to the longer breastfeeding periods among the children of homemakers. Furthermore, orphaned children who began receiving supplementary food immediately after birth were more likely to be underweight than those who started at six months or later. This is potentially due to their physiological inability to tolerate early supplementation [51].

## Limitations of the study

In this study there were observation bias, diet consumption recall bias, and a lack of study comparing children with families to those without families.

## Conclusion

In the current study, the prevalence of undernutrition in orphan children 6–59 month in Nekemte town was high. Vitamin A supplementation, source of child care, vaccination status, treatment for acute and chronic illness, age of the children, the occupation of the care giver, and the timing of the complimentary feeding initiation affected the orphaned children nutritional status. To address this, comprehensive approach that involves health care providers, care givers, and legal authorities is needed. In particular, regular vitamin A supplementation, timely vaccination, providing education to caregivers, regular health check-ups, nutritional assessments, and improved access to quality health care services could significantly reduce the prevalence of undernutrition among orphaned children.

## Supporting information

**Supporting Information 1.  English version of the Questionnaire.**
(DOCX)

**Supporting Information 2.  Dataset.**
(XLSX)

## Acknowledgments

We would like to express our sincere gratitude to Wollega University for their financial support and contribution to this study. We are also deeply thankful to the participants, the Nekemte Health Office, the data collectors, and the supervisors who dedicated their time and effort to make this research possible.

## Author contributions

**Conceptualization:** Werku Etafa, Fayera Abdana.

**Data curation:** Werku Etafa, Fayera Abdana, Emebet Bobo, Tesfaye Shibiru, Asefa Negeri, Dereje Temesgen.

**Formal analysis:** Werku Etafa, Fayera Abdana, Tesfaye Shibiru, Asefa Negeri, Gutu Leta, Dereje Temesgen.

**Funding acquisition:** Werku Etafa, Fayera Abdana, Asefa Negeri, Gutu Leta.

**Investigation:** Werku Etafa, Fayera Abdana, Wandimu Muche Mekonen, Meseret Belete Fite, Tesfaye Shibiru, Gutu Leta, Dereje Temesgen.

**Methodology:** Werku Etafa, Fayera Abdana, Wandimu Muche Mekonen, Dawit Tesfaye Daka, Meseret Belete Fite, Asefa Negeri, Dereje Temesgen.

**Project administration:** Werku Etafa, Fayera Abdana, Wandimu Muche Mekonen, Dawit Tesfaye Daka, Meseret Belete Fite, Tesfaye Shibiru, Gutu Leta, Dereje Temesgen.

**Resources:** Werku Etafa, Fayera Abdana, Emebet Bobo, Wandimu Muche Mekonen, Dawit Tesfaye Daka, Meseret Belete Fite, Tesfaye Shibiru, Dereje Temesgen.

**Software:** Werku Etafa, Fayera Abdana, Emebet Bobo, Wandimu Muche Mekonen, Dereje Temesgen.

**Supervision:** Werku Etafa, Fayera Abdana, Emebet Bobo, Wandimu Muche Mekonen, Dawit Tesfaye Daka, Meseret Belete Fite, Tesfaye Shibiru, Asefa Negeri, Gutu Leta, Dereje Temesgen.

**Validation:** Werku Etafa, Fayera Abdana, Emebet Bobo, Meseret Belete Fite, Tesfaye Shibiru, Gutu Leta.

**Visualization:** Werku Etafa, Fayera Abdana, Emebet Bobo, Dawit Tesfaye Daka.

**Writing – original draft:** Werku Etafa, Fayera Abdana, Wandimu Muche Mekonen, Dawit Tesfaye Daka, Asefa Negeri, Gutu Leta, Dereje Temesgen.

**Writing – review & editing:** Werku Etafa, Asefa Negeri, Gutu Leta.

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
