## [Decision Letter · Decision Letter 0]

21 Oct 2024

Dear Dr. Etafa,

Please submit your revised manuscript by Dec 05 2024 11:59PM. If you will need more time than this to complete your revisions, please reply to this message or contact the journal office at plosone@plos.org . A rebuttal letter that responds to each point raised by the academic editor and reviewer(s). You should upload this letter as a separate file labeled 'Response to Reviewers'.A marked-up copy of your manuscript that highlights changes made to the original version. You should upload this as a separate file labeled 'Revised Manuscript with Track Changes'.An unmarked version of your revised paper without tracked changes. You should upload this as a separate file labeled 'Manuscript'.

We look forward to receiving your revised manuscript.

Kind regards,

Chalachew Adugna Wubneh, MSc

Academic Editor

PLOS ONE

Journal Requirements:

1. When submitting your revision, we need you to address these additional requirements. Please ensure that your manuscript meets PLOS ONE's style requirements, including those for file naming. The PLOS ONE style templates can be found at https://journals.plos.org/plosone/s/file?id=wjVg/PLOSOne_formatting_sample_main_body.pdf and https://journals.plos.org/plosone/s/file?id=ba62/PLOSOne_formatting_sample_title_authors_affiliations.pdf 2. We note that your Data Availability Statement is currently as follows: [All relevant data are within the manuscript and its Supporting Information files.] Please confirm at this time whether or not your submission contains all raw data required to replicate the results of your study. Authors must share the “minimal data set” for their submission. PLOS defines the minimal data set to consist of the data required to replicate all study findings reported in the article, as well as related metadata and methods (https://journals.plos.org/plosone/s/data-availability#loc-minimal-data-set-definition). For example, authors should submit the following data: - The values behind the means, standard deviations and other measures reported;- The values used to build graphs;- The points extracted from images for analysis. Authors do not need to submit their entire data set if only a portion of the data was used in the reported study. If your submission does not contain these data, please either upload them as Supporting Information files or deposit them to a stable, public repository and provide us with the relevant URLs, DOIs, or accession numbers. For a list of recommended repositories, please see https://journals.plos.org/plosone/s/recommended-repositories. If there are ethical or legal restrictions on sharing a de-identified data set, please explain them in detail (e.g., data contain potentially sensitive information, data are owned by a third-party organization, etc.) and who has imposed them (e.g., an ethics committee). Please also provide contact information for a data access committee, ethics committee, or other institutional body to which data requests may be sent. If data are owned by a third party, please indicate how others may request data access. 3. Please include captions for your Supporting Information files at the end of your manuscript, and update any in-text citations to match accordingly. Please see our Supporting Information guidelines for more information: http://journals.plos.org/plosone/s/supporting-information.

Reviewers' comments:

Reviewer's Responses to Questions

**Comments to the Author**

1. Is the manuscript technically sound, and do the data support the conclusions?

Reviewer #1: Partly

Reviewer #2: Partly

2. Has the statistical analysis been performed appropriately and rigorously?

Reviewer #1: Yes

Reviewer #2: No

3. Have the authors made all data underlying the findings in their manuscript fully available?

Reviewer #1: Yes

Reviewer #2: Yes

4. Is the manuscript presented in an intelligible fashion and written in standard English?

Reviewer #1: No

Reviewer #2: Yes

Reviewer #1: I read the manuscript with interest. The manuscript has examined the Prevalence of undernutrition and its predictors among orphans under 5 years in Nekemte town, Ethiopia. My comments about this research have been listed below:

1. In the introduction section, line 88-91, the sentence is incomplete.

2. In the last paragraph of the introduction, the author claimed that there is no study investigating the prevalence of undernutrition in under-five orphan children in Nekemte town, but did not provide any information about the existing research in this area in other cities in Ethiopia, other countries or international data, if available. I found several similar studies conducted in Dilla (https://doi.org/10.1186/s40795-019-0295-6), Addis Ababa (https://doi.org/10.1186/s40795-021-00431-5), Gondar (10.11648/j.jfns.20140204.23), and Gambella Southwest (https://doi.org/10.1136/bmjopen-2020-045892) through a brief web search. Please review the similar studies in the introduction and provide the rationale for performing this study in Nekemte town.

3. In the method section, the sentence in which eligibility criteria have been listed (lines 109 -110) seems incorrect. It should be revised.

4. The author stated that the simple random sampling method has been applied. In this sampling method, we need a complete list of the target population members. It should be clarified where the researchers achieved such a list from. Are these data recorded in a registry system?

5. There is no information about the questionnaire used for assessing food security. Has the questionnaire been validated before in this population?  Is there a scoring system for this questionnaire in order to rate the food security status of the households or identify food insecure ones? Presenting the individual questions is not that informative. And the main question is that what was the aim of assessing food security status in this study?

6. In table 1, height and weight data have been presented. In the pediatric population raw anthropometric measures are not informative, please provide Z-score or percentile data.

7. in tables 1-4, descriptive demographic, environmental, social, etc., variables have been presented in a descriptive manner. I suggest to add two columns to these tables comparing the variables between well-nourished and malnourished children and use appropriate tests to explore the statistical significance of differences.  

7. In table 1, there are some repetitive rows, like “Occupation of the caretakers” or “Whom did the child lose”. Please revise them.

8. The second table in the text has been labeled as table 1 again.

9. The results of table 5 show that, not taking vitamin A supplements decrease the probability of being stunted. This is an unexpected result, and no explanation about that has been provided in the discussion section.

10. The whole manuscript text needs extensive English language editing.

Reviewer #2: Here are some comments that need to be addressed and focused on by the author regarding this manuscript:

1. Clarify the punctuation and eligibility criteria, with a focus on the specific target groups.

2. Provide more details on sample size determination, including specifications for the non-response rate.

3. Ensure appropriate referencing for the operational definitions of stunting, wasting, and other terms.

4. The report lacks proper definitions or explanations of the abbreviations used, such as FGFS.

5. The interpretation of the analysis needs to be addressed more thoroughly. In some cases, interpretations are provided without corresponding information in any table. Recheck the analysis to identify whether additional information should be included or if irrelevant statistics need to be removed.

6. Tables in the manuscript need to be properly formatted with accurate headers and estimates.

7. In the results section, there are many variables presented in the tables that are not interpreted or discussed. This should be reviewed for completeness.

8. The analysis may need corrections. Basic frequency tables should be rechecked, and there are significant issues with skip-pattern questions that require attention.

9. The manuscript needs to be formatted in accordance with the journal's style, with proper table and figure names, as well as accurate notations in the text for references.

10. The paper requires improvements in the interpretation of results and the presentation of tables. The interpretation of odds ratios is unclear and needs more careful review to ensure accuracy throughout the document.

11. The discussion needs to be improved based on the revised results, along with the existing scenarios.

12. A revised version is required with corrected analysis, accurate interpretation, and potential restructuring of sections

**Do you want your identity to be public for this peer review?** For information about this choice, including consent withdrawal, please see our Privacy Policy

Reviewer #1: No

Reviewer #2: **Yes: ** Arifa Tabassum

---

## [Author Response · Author response to Decision Letter 1]

3 Apr 2025

Responses to Reviewers

Dear Editor,

Thank you for your patience and for the valuable scientific contributions you made to enhance our manuscript's readability and credibility. We have addressed the comments and questions raised by the reviewers in detail below.

Response to Editor,

Comments: Please ensure that your manuscript meets PLOS ONE's style requirements, including those for file naming.

Response: Thank you for suggesting to rearrange. The revised manuscript is rearranged according to the manuscript. This can be seen throughout the document.

Comment: Is the manuscript technically sound, and do the data support the conclusions?

Response: We modified the conclusions as suggested and stated it as below (page 2, line 57-60).

Conclusions: The study concludes that nursing students have limited knowledge and skills regarding children's environmental health. To address this gap, it is strongly recommended to integrate children's environmental health into the nursing curriculum and to encourage nursing students to enhance their understanding and improve their skills in this vital area.

Comment: Has the statistical analysis been performed appropriately and rigorously?

Response: For this questions there was no question raised. Thus, we tried to identify mistakes in the statistical part. However, we didn’t make any changes to the statistical parts as there is noted mistake to our level of checking.

Comment: Have the authors made all data underlying the findings in their manuscript fully available?

Response: The findings tried to answer the objectives of the study and are included in the results sections.

Comment: Is the manuscript presented in an intelligible fashion and written in standard English?

Response: We tried to improve the standard of the English language usage. This could be seen throughout the document.

Dear Reviewers,

Thank you for your valuable insights that have significantly improved the quality of our manuscript. We appreciate the time you dedicated to this process and your contributions to the scientific community, benefiting people around the globe. Below are our responses to the comments and questions raised.

Response to Reviewer #1

1. Thank you for your great emphasis, we left to add ‘were associated with undernutrition in under five year’s old children”. Therefore, we have corrected as the following Evidence revealed that educational level, lack of employment for the head of the household, large family size, and food insecurity, insufficient food consumption, and a high rate of infectious illnesses, lack of vitamin A supplementation, poor knowledge and attitude of caregivers, and diarrhea were associated with undernutrition in under five years old orphan children.

2. This is very important comment that enhances us to see many sources related to our article. But, as you mentioned, we only focused on the Nekemte city which is our main study area in context of the topic. We have included in the revise version of manuscript

3. Exactly, it was a typing error. We wanted to say, orphan children 6-59 month living in Nekemte Town who are paired with caregivers or guardians were included and orphan children who were who were seriously ill and with physical disabilities as well as whose caregivers had hearing loss were excluded from the study. So, we have corrected on the manuscript with highlight text.

4. Thank you very much, we assumed as if the one displayed in figure is enough to yield full information. Now we have added to the manuscript in both text and figure according to following. The number of under-five orphan children in Nekemte town is 663; out of 295 are male, and 368 are female. Sample size was calculated to get 384 as final. Data was taken from orphan children registration book in Nekemte town children, women, and youth affairs Bureau.

5. Best reminder; we have not conducted food security which is to be another study. For the moment, we only focused on the locally available food and traditional eating behavior. Other questionnaires were pretested before actual data collection that kept the quality of information

6. Thank you a lot, we have corrected in the revised manuscript

7. Good idea, but, we only described the magnitude of each data in each tables. For the statistical significance of variables, we used table 5-6 as described in manuscript. If it is not clear, we are here to listen your second comment with apologize. It is reverent comment. We used the two phrases for different information. For example,” Occupation of the caretakers” speaks about what do the care takers do as regular activity to survive. The second “Whom did the child lose” wants to say from the parent, did a father or a mother die exposing them be orphan? So, let it be understood in such manner.

8. Exactly, thank you really. That was a typing error again, we have corrected it now in revised manuscript

9. Very impressive comment. The result was mistakenly written in front of Not, but it should be on Yes line. Now we have corrected on revised manuscript

10. Thank you very much for amazing comments. We will correct each and every comments in revised manuscript.

Responses to Reviewer #2

Thank you a lot. We have revised the punctuation and grammar error in manuscript that will see it now. On eligibility criteria we have corrected as “orphan children 6-59 month living in Nekemte Town who are paired with caregivers or guardians were included and orphan children who were who were seriously ill and with physical disabilities as well as whose caregivers had hearing loss were excluded from the study”.

1. You reminded as important point. The sample size was calculated for both first and second objectives based on available references. Accordingly, we have calculated for the first objective using a study done in Addis Ababa City (Teferi H, 2021) on prevalence of stunting, wasting and underweight that yield 349 as maximum from stunting prevalence compared to others. For the second objective, we used same article showing different significant variables that yield 334 as a maximum sample size based on Vitamin A supplementation. Therefore, from both sample size, 349 taken as the maximum and 10% non-response rate was added to be the final 384 as described in figure in revised form of manuscript.

2. Okay, thank you! Now we have corrected in revised manuscript

3. Nice reminder, we crossed over it while searching abbreviations in the body texts. We mean to say “Fibroblast Growth Factors” for FGFs.

4. Thank you very much! But, it is good if you as any specific areas we poorly interpreted to reach at it as easy as possible. Anyhow, we will try to cross check all parts.

5. Yes, we mistakenly made some error on typing; now we have corrected in revised manuscript.

6. Thank you. But, again the area where we have to give consideration is better to be shown us specifically to correct accordingly. We hope, we did not leave to mention any significant variable except insignificant one which is not this much important. We see it again.

7. Very important comment, now we have corrected in the revised manuscript that you will come across.

8. Following the journal guideline is mandatory as you exactly pressed. We tried to submit the manuscript in texts, table and figure separately. The remained will be corrected again.

9. Important point, we will do on it.

10. Thank you again we tried to revise based on the given comments and our reviews. If we miss any part, we are ready to accept your feedback more.

---

## [Editor Report · Decision Letter 1]

22 Apr 2025

Dear Dr. Etafa,

Thank you for submitting your manuscript to PLOS ONE. After careful consideration, we feel that it has merit but does not fully meet PLOS ONE’s publication criteria as it currently stands. Therefore, we invite you to submit a revised version of the manuscript that addresses the points raised during the review process.

We look forward to receiving your revised manuscript.

Kind regards,

Chalachew Adugna Wubneh, MSc

Academic Editor

PLOS ONE

Additional Editor Comments:

Dear Authors

Thank you for your revision and re-submission, but specially your response to the editor part is not properly responded, could you recheck the documents and all responses and resubmit?

Thank you.

---

## [Author Response · Author response to Decision Letter 2]

23 Apr 2025

Response to Editor

Dear Editor,

Thank you for your patience and for the valuable scientific contributions you made to enhance our manuscript's readability and credibility. We have addressed the comments and questions raised by the editor in detail below.

Response to Editor,

Dear Editor, we appreciate your commitment and excused for the delay.

Comment: Please include a rebuttal letter that responds to each point raised by the academic editor and reviewer(s). You should upload this letter as a separate file labeled 'Response to Reviewers'.

Response: Thank you very much. We have provided answers to each questions raised line by line which can be evidenced from the cleaned version of the manuscript. The lines and pages stated in this responses are those in the cleaned version of the manuscript.

Comment: A marked-up copy of your manuscript that highlights changes made to the original version. You should upload this as a separate file labeled 'Revised Manuscript with Track Changes'.

Response: A cleaned version of the manuscript is uploaded in the system.

Comment: An unmarked version of your revised paper without tracked changes. You should upload this as a separate file labeled 'Manuscript'.

Response: A tracked version of the manuscript is uploaded and found in the system.

Comment: Comments: When submitting your revision, address these additional requirements.

Please ensure that your manuscript meets PLOS ONE's style requirements, including those for file naming. The PLOS ONE style templates can be found at https://journals.plos.org/plosone/s/file?id=wjVg/PLOSOne_formatting_sample_main_body.pdf and

Response: Thank you for suggesting to rearrange. The revised manuscript is rearranged according to the manuscript. This can be seen throughout the document.

Comment: We note that your Data Availability Statement is currently as follows: [All relevant data are within the manuscript and its Supporting Information files. Please confirm at this time whether or not your submission contains all raw data required to replicate the results of your study. Authors must share the “minimal data set” for their submission. PLOS defines the minimal data set to consist of the data required to replicate all study findings reported in the article, as well as related metadata and methods (https://journals.plos.org/plosone/s/data-availability#loc-minimal-data-set-definition). For example, authors should submit the following data:

• The values behind the means, standard deviations and other measures reported;

• The values used to build graphs;

• The points extracted from images for analysis.

Response: We apologize for any confusion. The dataset referenced in this manuscript is submitted alongside the revised manuscript. The values behind means, standard deviations and other reports, graph values and images reported in the manuscript are based on the data in the data set. It is provided in an Excel spreadsheet format, as we were informed that the minimum dataset submitted in SPSS could not be opened. This dataset has been uploaded to the system during the submission of the revised work. The age, residence and date of birth of the study participants were removed from the data set. Otherwise, other relevant data are found in the manuscript.

Authors’ responses to journal questions

Comment: Is the manuscript technically sound, and do the data support the conclusions?

Response: We modified the conclusions as suggested and stated it as below (page 2, line 57-60). In this section, the minor technical error was made based on the given suggestion. It is rewritten as follow:

“Conclusions: The study concludes that nursing students have limited knowledge and skills regarding children's environmental health. To address this gap, it is strongly recommended to integrate children's environmental health into the nursing curriculum and to encourage nursing students to enhance their understanding and improve their skills in this vital area”.

Comment: Has the statistical analysis been performed appropriately and rigorously?

Response: For this questions there was no question raised. Thus, we tried to identify mistakes in the statistical part. However, we didn’t make any changes to the statistical parts as there is noted mistake to our level of checking.

Comment: Have the authors made all data underlying the findings in their manuscript fully available?

Response: The findings tried to answer the objectives of the study and are included in the results sections, and discussed further in the discussion section.

Comment: Is the manuscript presented in an intelligible fashion and written in Standard English?

Response: We tried to improve the standard of the English language usage. This could be seen throughout the document.

Dear Reviewers,

Thank you for your valuable insights that have significantly improved the quality of our manuscript. We appreciate the time you dedicated to this process and your contributions to the scientific community, benefiting people around the globe. Below are our responses to the comments and questions raised.

Response to Reviewer #1

Comment: In the introduction section, line 88-91, the sentence is incomplete.

Response: Thank you for your great emphasis, we left to add ‘were associated with undernutrition in under five year’s old children”. Therefore, we have corrected as the following Evidence revealed that educational level, lack of employment for the head of the household, large family size, and food insecurity, insufficient food consumption, and a high rate of infectious illnesses, lack of vitamin A supplementation, poor knowledge and attitude of caregivers, and diarrhea were associated with undernutrition in under five years old orphan children (page 4, line: 101-108).

Comment: In the last paragraph of the introduction, the author claimed that there is no study investigating the prevalence of undernutrition in under-five orphan children in Nekemte town, but did not provide any information about the existing research in this area in other cities in Ethiopia, other countries or international data, if available. I found several similar studies conducted in Dilla (https://doi.org/10.1186/s40795-019-0295-6), Addis Ababa (https://doi.org/10.1186/s40795-021-00431-5), Gondar (10.11648/j.jfns.20140204.23), and Gambella Southwest (https://doi.org/10.1136/bmjopen-2020-045892) through a brief web search. Please review the similar studies in the introduction and provide the rationale for performing this study in Nekemte town.

Response: This is very important comment that enhances us to see many sources related to our article. But, as you mentioned, we only focused on the Nekemte city which is our main study area in context of the topic. We have included in the revise version of manuscript, specifically in the statement of the problem and discussion sections.

Comment: In the method section, the sentence in which eligibility criteria have been listed (lines 109 -110) seems incorrect. It should be revised.

Response: Exactly, it was a typing error. We wanted to say, orphan children 6-59 month living in Nekemte Town who are paired with caregivers or guardians were included and orphan children who were who were seriously ill and with physical disabilities as well as whose caregivers had hearing loss were excluded from the study. So, we have corrected on the manuscript with highlight text.

Comment: The author stated that the simple random sampling method has been applied. In this sampling method, we need a complete list of the target population members. It should be clarified where the researchers achieved such a list from. Are these data recorded in a registry system?

Response: The study participants were selected using simple random sampling techniques. We employed a lottery method based on the registration books to generate a sampling frame. The registry was obtained from the town youth and women' office. When two or more orphans were discovered in the same household, the younger one was chosen for the study (page 5, line: 141-144).

Comment: There is no information about the questionnaire used for assessing food security. Has the questionnaire been validated before in this population? Is there a scoring system for this questionnaire in order to rate the food security status of the households or identify food insecure ones? Presenting the individual questions is not that informative. And the main question is that what was the aim of assessing food security status in this study?

Response: Thank you very much, we assumed as if the one displayed in figure is enough to yield full information. Now we have added to the manuscript in both text and figure according to following. The number of under-five orphan children in Nekemte town is 663; out of 295 are male, and 368 are female. Sample size was calculated to get 384 as final. Data was taken from orphan children registration book in Nekemte town children, women, and youth affairs Bureau (page 5, line: 130-139).

Comment: In table 1, height and weight data have been presented. In the pediatric population raw anthropometric measures are not informative, please provide Z-score or percentile data

Response: Best reminder; we have not conducted food security which is to be another study. For the moment, we only focused on the locally available food and traditional eating behavior. Other questionnaires were pretested before actual data collection that kept the quality of information (Table1, page 10).

Comment: In tables 1-4, descriptive demographic, environmental, social, etc., variables have been presented in a descriptive manner. I suggest to add two columns to these tables comparing the variables between well-nourished and malnourished children and use appropriate tests to explore the statistical significance of differences.

Response: Thank you for relevant information. We did not use in the table because of our dependent variables are categorized into four: normal, underweight, wasting and stunting. Comment: In table 1, there are some repetitive rows, like “Occupation of the caretakers” or “Whom did the child lose”. Please revise them.

Response: We used the two phrases for different information. For example,” Occupation of the caretakers” speaks about what do the care takers do as regular activity to survive. The second “Whom did the child lose” wants to say from the parent, did a father or a mother die exposing them be orphan? So, let it be understood in such manner.

Comment: The second table in the text has been labeled as table 1 again.

Response: Exactly, thank you really. That was a typing error again, we have corrected it now in revised manuscript

Comment: The results of table 5 show that, not taking vitamin A supplements decrease the probability of being stunted. This is an unexpected result, and no explanation about that has been provided in the discussion section.

Response: Thank you very much. It was rewritten as “The study found that orphans who did not receive vitamin A supplements had a 41% higher chance of developing stunting than those who did (AOR: 0.59, 95% CI: 0.371-0.94)” (PAHE 15, LINE: 281 & 282). Absence of the vitamin A exposes the child for stunting.

Comment: The whole manuscript text needs extensive English language editing

Response: We tried to reach for the English language accuracy.

Responses to Reviewer #2

Comment: Clarify the punctuation and eligibility criteria, with a focus on the specific target groups.

Response: Thank you a lot. We have revised the punctuation and grammar error in manuscript that will see it now. On eligibility criteria we have corrected as “orphan children 6-59 month living in Nekemte Town who are paired with caregivers or guardians were included and orphan children who were who were seriously ill and with physical disabilities as well as whose caregivers had hearing loss were excluded from the study”.

Comment: Provide more details on sample size determination, including specifications for the non-response rate.

Response: You reminded as important point. The sample size was calculated for both first and second objectives based on available references. Accordingly, we have calculated for the first objective using a study done in Addis Ababa City (Teferi H, 2021) on prevalence of stunting, wasting and underweight that yield 349 as maximum from stunting prevalence compared to others. For the second objective, we used same article showing different significant variables that yield 334 as a maximum sample size based on Vitamin A supplementation. Therefore, from both sample size, 349 taken as the maximum and 10% non-response rate was added to be the final 384 as described in figure in revised form of manuscript.

Comment: Ensure appropriate referencing for the operational definitions of stunting, wasting, and other terms.

Response: Okay, thank you! Now we have corrected in revised manuscript

Comment: The report lacks proper definitions or explanations of the abbreviations used, such as FGFS.

Response: Nice reminder, we crossed over it while searching abbreviations in the body texts. We mean to say “Fibroblast Growth Factors” for FGFs.

Response: Thank you very much! But, it is good if you as any specific areas we poorly interpreted to reach at it as easy as possible. Anyhow, we will try to cross check all parts.

Comment: The interpretation of the analysis needs to be addressed more thoroughly. In some cases, interpretations are provided without corresponding information in any table. Recheck the analysis to identify whether additional information should be included or if irrelevant statistics need to be removed

Response: Yes, we mistakenly made some error on typing; now we have corrected in revised manuscript.

Comment: Tables in the manuscript need to be properly formatted with accurate headers and estimates.

Response: Thank you. We modified in the revised manuscript.

Comment: In the results section, there are many variables presented in the tables that are not interpreted or discussed. This should be reviewed for completeness

Response: Essential. Now we have corrected in the revised manuscript that you will come across.

Comment: Revise manuscript according the journal style, English language usage and discussion reconstruction

Response: We tried to improve our drawbacks which could be seen from the revised manuscript.. We tried to submit the manuscript in texts, table and figure separately. The remained will be corrected again. Thank you again we tried to revise based on the given comments and our reviews. If we miss any part, we are ready to accept your feedback more.

---

## [Decision Letter · Decision Letter 2]

11 Jun 2025

Dear Dr. Etafa,

Thank you for submitting your manuscript to PLOS ONE. After careful consideration, we feel that it has merit but does not fully meet PLOS ONE’s publication criteria as it currently stands. Therefore, we invite you to submit a revised version of the manuscript that addresses the points raised during the review process.

We look forward to receiving your revised manuscript.

Kind regards,

Chalachew Adugna Wubneh, MSc

Academic Editor

PLOS ONE

**Journal Requirements:**

**Additional Editor Comments:**

Dear Authors

Pleas kindly address all the comments and submit the revised version.

Specifically, you have to address the sample size determination issues clearly considering the three primary outcomes. In addition. I am not clear with 348 and 384?

Editorial comments,

1. Abstract line 54 U5C, is not standard form of writing, write fully and even in the main document.

2. Operationalized orphan in your study context

3. Editorial issues from line 240 and 252, 254, 265

4. In the factor table-5 sub-section you mentioned wasting two times and you missed underweight part

5. How do you approach the data collection process, is that school, community house to house or other settings clearly state in the method part

Reviewers' comments:

Reviewer's Responses to Questions

**Comments to the Author**

Reviewer #2: All comments have been addressed

Reviewer #3: (No Response)

2. Is the manuscript technically sound, and do the data support the conclusions?

Reviewer #2: Yes

Reviewer #3: Yes

3. Has the statistical analysis been performed appropriately and rigorously?

Reviewer #2: No

Reviewer #3: Yes

4. Have the authors made all data underlying the findings in their manuscript fully available?

Reviewer #2: Yes

Reviewer #3: Yes

5. Is the manuscript presented in an intelligible fashion and written in standard English?

Reviewer #2: No

Reviewer #3: Yes

**Reviewer #2:**  1.Sample size inconsistency:

The manuscript states that the sample size was determined to be 384, accounting for a non-response rate, yet the final number of participants used in the analysis was 373. Please provide a clear explanation of the sample size calculation and the reason for this discrepancy. This is a critical methodological component of the study and must be transparent.

2.Statistical notation consistency:

Please maintain consistency in reporting descriptive statistics. Use the format ‘Mean ± SD’ instead of ‘Mean and SD’. Also, when reporting frequencies, please use the format ‘Frequency (n)’ to improve clarity and consistency.

3.Unclear software-based reference before table:

There is a reference like “Error! Reference source not found.” before Table 1. This appears to be a software-generated error, likely from a broken cross-reference in Word. Please clarify what was originally intended to be communicated at that location or remove it if not relevant.

4.Table 2 format issues (WFA, HFA, WFH):

The structure of Table 2 requires correction, particularly in the presentation of WFA, HFA, and WFH. Please ensure the variables are clearly defined, appropriately labeled, and aligned with standard anthropometric reporting formats.

5.Table mislabeling:

Table 2 is mistakenly labeled as Table 1. Please correct the numbering to maintain logical sequence throughout the manuscript.

6.Percentage total inconsistencies (Table 4):

Percentages must always sum to 100% within their respective categories. For follow-up questions that apply only to a subset of respondents (i.e., those who answered ‘Yes’), ensure that the “n” reflects only that subgroup rather than the total sample.

7. Missing frequency data for stunting and wasting in Table 1:

Frequencies for stunting and wasting are not included in Table 1. As these are key outcome variables, please ensure that their frequencies are reported clearly alongside their corresponding percentages.

**Reviewer #3:**  Thanks for the opportunity to review this manuscript. The authors have made some important revisions based on the two previous reviewers’ comments and the content has been strengthened. Having reviewed the revised version, I have just a few comments:

1. The study presents the results of original research.

Yes, this study presents results from an original cross-sectional research study.

2. Results reported have not been published elsewhere.

It appears that the study results have not been published in another peer-reviewed journal.

3. Experiments, statistics, and other analyses are performed to a high technical standard and are described in sufficient detail.

• Table 1: Socio-Demographic Characteristics of Orphans and Their Caregivers in Nekemte Town,Ethiopia, 2023--- provide the upper age limit in months for the second child age category (≥24 months)

• Table 1: Health status, environmental, behavioral, social and legal services, knowledge and attitude related characteristics of orphan children in Nekemte town, Ethiopia, 2023--- This table should be labelled at Table #2. In addition, the title refers to “attitude-related” characteristics but I did not see any data in this table related to attitudes. Please explain, revise as appropriate.

• Table 3—explain what is meant by “flat foods”

• Please review and edit for clarity the track changes for the summary of results in the discussion section in lines 354 to 360 as Addis Adaba is mentioned multiple times.

4. Conclusions are presented in an appropriate fashion and are supported by the data.

• The data presented in the manuscript supports the conclusions.

• Some suggestions in the conclusion section go beyond the data collected in this study. For example, the reasons that some guardians lack knowledge of proper child feeding practices is not clear based on the study data presented. There is a suggest by the authors to improve the knowledge and communications skills of health workers but it is not clear if their knowledge/skills are lacking or if guardians are not having sufficient contacts with health workers for those workers to convey their knowledge. The fact that under-immunization is a problem in the study population suggests that access to/utilization of care is an issue, More information would be needed beyond the scope of what was collected in this study to understand the level of health care worker knowledge of child nutrition and feeding practices.

• Lines 463-464 in the Conclusion section should be edited for clarity and grammar (e.g., change “frequency to prevalence” and revise “marked public health significance”: Conclusion

463 Overall, the frequency of stunting, wasting, and underweight among orphaned children was

464 marked public health significance.

4. The article is presented in an intelligible fashion and is written in standard English.

The article is written in standard English that is acceptable. There are some sentences and definitions that lack clarity. For example:

• The introductions says: “According to Ethiopia's malnutrition trend, the prevalence of stunting has reduced from 58% to 38% and that of underweight has fallen from 41% to 24% over the last fifteen 15 years from 2000. However, the prevalence of wasting has marginally declined during the previous 15 years, from 12% to 10% (4,

75 5).” When referring to the previous 15 years, the reader would anticipate this time period would be between 2010 and 2025. However, these two sentences appear to refer to 2000-2015.

• The definitions of food security and insecurity in the Methods section do not appear to conform with global standardized definitions. For example, they are not aligned with the FAO definitions below:

o Food security: a situation that exists when all people, at all times, have physical, social and economic access to sufficient, safe and nutritious food that meets their dietary needs and food preferences for an active and healthy life.

o A person is food insecure when they lack regular access to enough safe and nutritious food for normal growth and development and an active and healthy life. This may be due to unavailability of food and/or lack of resources to obtain food

Or the FANTA definition for food security:

• Food security in a population means that all people, at all times, have sufficient access to food to meet their dietary needs for a productive and healthy life.

Suggest further clarifying the language in the definitions used in this study, including the source of the definitions being used.

6. The research meets all applicable standards for the ethics of experimentation and research integrity.

The research appears to meet the required ethical standards. This study involved human subjects and the authors obtained ethical approval from the Wollega University Research Ethics Committee.

**Do you want your identity to be public for this peer review?** For information about this choice, including consent withdrawal, please see our Privacy Policy

Reviewer #2: No

Reviewer #3: No

---

## [Author Response · Author response to Decision Letter 3]

9 Jul 2025

Responses to Editors and Reviewers

Dear Editor and reviewers,

Thank you very much for sharing your invaluable resources with us: your time, deep knowledge, and expertise, which have greatly enhanced our understanding of undernutrition in orphans aged 6 to 59 months. We sincerely appreciate the significant effort you invested in shaping our manuscript to improve its quality. We have learned a great deal from your comments and have made corrections in the areas identified for improvement. We are grateful for your insights and have accepted the suggestions provided.

Editors’ comment

Dear Authors, Please kindly address all the comments and submit the revised version. Specifically, you have to address the sample size determination issues clearly considering the three primary outcomes. In addition. I am not clear with 348 and 384?

Response: Thank you very much for great concern. It is a clear mistake. We corrected it as follow: We estimated the sample size using the single population proportion based on a 5% margin of error, a 95% confidence interval and the prevalence of stunting from the study conducted in Addis Ababa (34.8%) among orphans aged 6-59-months(9). Based on these assumptions, we obtained 349 samples. By adding a 10% for the nonresponse rates, the final sample size was 384 (page 9, line 130-133).

Editorial comments

Comment 1: Abstract line 54 U5C, is not standard form of writing, write fully and even in the main document.

Response: Thank you. We removed from this section also suggested by reviewer #3.

Comment 2: Operationalized orphan in your study context

Response: Thank you. We provided the definition of an orphan in the revised manuscript under subheading of operational definitions as “Orphan: is a child who has lost one or both parents through death (Page 6, line 144).

Comment 3. Editorial issues from line 240 and 252, 254, 2654. In the factor table-5 sub-section you mentioned wasting two times and you missed underweight part5.

Response: Thank you very much. These parts are corrected throughout the document.

Comment 4: How do you approach the data collection process, is that school, community house to house or other settings clearly state in the method part

Response: The study was community based. We introduced in the methods section under study setting, population and period. We apologize for not mentioning it in the methods section.

Responses to Reviewer #2:

Comment 1: Sample size inconsistency: The manuscript states that the sample size was determined to be 384, accounting for a non-response rate, yet the final number of participants used in the analysis was 373. Please provide a clear explanation of the sample size calculation and the reason for this discrepancy. This is a critical methodological component of the study and must be transparent.

Response: Thank you very much for great concern. It is a clear mistake. We corrected it as follow:

We estimated the sample size using the single population proportion based on a 5% margin of error, a 95% confidence interval and the prevalence of stunting from the study conducted in Addis Ababa (34.8%) among orphans aged 6-59-months(9). Based on these assumptions, we obtained 349 samples. By adding a 10% for the nonresponse rates, the final sample size was 384 (page 9, line 130-133).

Comment 2: Statistical notation consistency: Please maintain consistency in reporting descriptive statistics. Use the format ‘Mean ± SD’ instead of ‘Mean and SD’. Also, when reporting frequencies, please use the format ‘Frequency (n)’ to improve clarity and consistency.

Response: We reorganized it as follow: The mean and standard deviation (Mean ±SD) of age, weight and height of the orphans was 33.78±17.8 SD months, 12.99±3.25 kg, and 88.22 ± 14.83cm, respectively. Response: We modified the points in the revised manuscript (page 9, line232-240).

Comment 3: Unclear software-based reference before table: There is a reference like “Error! Reference source not found.” before Table 1. This appears to be a software-generated location or remove it if not relevant.

Response: Great. It is modified using the system (Page 9, line 240-242).

Comment 4: Table 2 format issues (WFA, HFA, WFH): The structure of Table 2 requires correction, particularly in the presentation of WFA, HFA, and WFH. Please ensure the variables are clearly defined, appropriately labeled, and aligned with standard anthropometric reporting formats.

Response: Thank you. The dependent variables (underweight, height and weight) are redefined in the methods section under the subheading operational definitions. The abbreviated variables WFA, HFA, and WFH in the table one are removed as it is similar to the result findings in the figure 1(page 9, table1 and page 14, 275-278).

Comment 5: Table mislabeling: Table 2 is mistakenly labeled as Table 1. Please correct the numbering to maintain logical sequence throughout the manuscript.

Response: We modified this table using the system generated and correctly ordered in the revised version of the manuscript (page 11).

6. Percentage total inconsistencies (Table 4): Percentages must always sum to 100% within their respective categories. For follow-up questions that apply only to a subset of respondents (i.e., those who answered ‘Yes’), ensure that the “n” reflects only that subgroup rather than the total sample.

Response: Thank you very much for your suggestion. We made several corrections regarding the frequencies and percentage for table 4 as suggested (Table 4, page 12 and 13).

7. Missing frequency data for stunting and wasting in Table 1: Frequencies for stunting and wasting are not included in Table 1. As these are key outcome variables, please ensure that their frequencies are reported clearly alongside their corresponding percentages.

Response: Thank you very much. It was wrongly stated. However, since the correct one is mentioned under the subtopic prevalence in figure one, we prefer to use it (Page 14, line 275-278).

Responses to Reviewer#3

Comment 1: Socio-Demographic Characteristics of Orphans and Their Caregivers in Nekemte Town, Ethiopia, 2023--- provide the upper age limit in months for the second child age category (≥24 months).

Response: Thank you for your informative comment. We takes a look at and edited as you suggested throughout the tables and documents.

Comment 2: Health status, environmental, behavioral, social and legal services, knowledge and attitude related characteristics of orphan children in Nekemte town, Ethiopia, 2023--- This table should be labelled at Table #2. In addition, the title refers to “attitude-related” characteristics but I did not see any data in this table related to attitudes. Please explain, revise as appropriate.

Response: Thank you. We revised using the table labels using the insert caption. However, the attitude of the caregivers typed error. In the revised version of our manuscript, we have removed it (Table 2, Page 11& 12).

Comment 3: Table 3—explain what is meant by “flat foods”

Response. Thank you very much. It is typing error and corrected to ‘fat’ (Table3, page 12 &13).

Comment 4: Please review and edit for clarity the track changes for the summary of results in the discussion section in lines 354 to 360 as Addis Ababa is mentioned multiple times.

Response: Alright. There is a repetition. It modified in its revised version of the manuscript (page, line).

Comment 5: Some suggestions in the conclusion section go beyond the data collected in this study. Lines 463-464 in the Conclusion section should be edited for clarity and grammar (e.g., change “frequency to prevalence” and revise “marked public health significance”: Conclusion: Overall, the frequency of stunting, wasting, and underweight among orphaned children was marked public health significance

Response: It is revised as follow “It is crucial to enhance the knowledge and communication skills of healthcare workers and caregivers regarding infant and young child feeding practices and vitamin A supplementation for children under five years of age. Additionally, improving supervision of orphans by legal bodies is essential for better health outcomes.” (page 2, line 52-56).

Comment 6: The article is presented in an intelligible fashion and is written in Standard English. The article is written in Standard English that is acceptable. There are some sentences and definitions that lack clarity. For example:• The introductions says: “According to Ethiopia's malnutrition trend, the prevalence of stunting has reduced from 58% to 38% and that of underweight has fallen from 41% to 24% over the last fifteen 15 years from 2000. However, the prevalence of wasting has marginally declined during the previous 15 years, from 12% to 10% (4,75 5).” When referring to the previous 15 years, the reader would anticipate this time period would be between 2010 and 2025. However, these two sentences appear to refer to 2000-2015.

Response: Thank you very much. We corrected 15 years to 2000-2015 years (page 3, line 70-72).

Comment 7: The definitions of food security and insecurity in the Methods section do not appear to conform to global standardized definitions. For example, they are not aligned with the FAO or FANTA definitions.

Response: Thank you very much. We revised the definition of food security and insecurity by replacing the operational definition suggested for this study and cited (page 6, line 157-162).

---

## [Decision Letter · Decision Letter 3]

19 Aug 2025

Dear Dr. Etafa,

Thank you for submitting your manuscript to PLOS ONE. After careful consideration, we feel that it has merit but does not fully meet PLOS ONE’s publication criteria as it currently stands. Therefore, we invite you to submit a revised version of the manuscript that addresses the points raised during the review process.

**ACADEMIC EDITOR: **

We look forward to receiving your revised manuscript.

Kind regards,

Chalachew Adugna Wubneh, MSc

Academic Editor

PLOS ONE

Journal Requirements:

Reviewers' comments:

Reviewer's Responses to Questions

**Comments to the Author**

Reviewer #2: All comments have been addressed

Reviewer #3: All comments have been addressed

2. Is the manuscript technically sound, and do the data support the conclusions?

Reviewer #2: Yes

Reviewer #3: (No Response)

3. Has the statistical analysis been performed appropriately and rigorously?

Reviewer #2: No

Reviewer #3: (No Response)

4. Have the authors made all data underlying the findings in their manuscript fully available?

Reviewer #2: Yes

Reviewer #3: (No Response)

5. Is the manuscript presented in an intelligible fashion and written in standard English?

Reviewer #2: No

Reviewer #3: (No Response)

Reviewer #2: Great correction, I appreciate it. However,

1. In Table 4, for the question 'Did you worry about your family not having enough food?', there is a skip pattern issue that has not been corrected accordingly.

2. The other corrections seem fine, but there are still some spacing inconsistencies throughout the manuscript. Please review and correct these carefully.

Reviewer #3: (No Response)

**Do you want your identity to be public for this peer review?** For information about this choice, including consent withdrawal, please see our Privacy Policy

Reviewer #2: No

Reviewer #3: No

---

## [Author Response · Author response to Decision Letter 4]

26 Sep 2025

Response to Editor and Reviewers

Dear Editor and Reviewers,

Thank you for your time and the valuable scientific contributions you made to improve the quality of our manuscript. We appreciate your insights and suggestions, which have significantly enhanced our work.

Comment: A rebuttal letter that responds to each point raised by the academic editor and reviewer(s). You should upload this letter as a separate file labeled 'Response to Reviewers'.

Response: A letter responds to each point raised raised by the academic editor and reviewers is provided and attached in the online submission with the file name labelled as ’Response to Reviewers.

Comment: A marked-up copy of your manuscript that highlights changes made to the original version. You should upload this as a separate file labeled 'Revised Manuscript with Track Changes'.

Response: A manuscript with track version is submitted with the file name labelled as 'Revised Manuscript with Track Changes'.

Comment: An unmarked version of your revised paper without tracked changes. You should upload this as a separate file labeled 'Manuscript'.

Response: Cleaned version of the manuscript is submitted with the file name labelled as ‘Manuscript’.

Journal Requirements:

Comment: If the reviewer comments include a recommendation to cite specific previously published works, please review and evaluate these publications to determine whether they are relevant and should be cited. There is no requirement to cite these works unless the editor has indicated otherwise.

Response: We did not encounter this kind of recommendation.

Comment: Please review your reference list to ensure that it is complete and correct. If you have cited papers that have been retracted, please include the rationale for doing so in the manuscript text, or remove these references and replace them with relevant current references. Any changes to the reference list should be mentioned in the rebuttal letter that accompanies your revised manuscript. If you need to cite a retracted article, indicate the article’s retracted status in the References list and also include a citation and full reference for the retraction notice.

Response: "We did not include any retracted papers in this manuscript."

Reviewers Comments

Comment: Great correction, I appreciate it. However1. In Table 4, for the question 'Did you worry about your family not having enough food?', there is a skip pattern issue that has not been corrected accordingly.

Response: Thank you very much for your attention, as it is greatly appreciated. We identified an error, which we corrected by calculating the percentage of the 190 respondents who answered 'yes' to the question, 'Did you worry about your family not having enough food?' (see Table 4).

Comment: The other corrections seem fine, but there are still some spacing inconsistencies throughout the manuscript. Please review and correct these carefully.

Response: We have identified several inconsistencies in our manuscript and have made the necessary corrections. This is found throughout the manuscript.

---

## [Decision Letter · Decision Letter 4]

24 Oct 2025

Prevalence of undernutrition and its associated factors among orphans aged 6-59 months in Nekemte town, Ethiopia

PONE-D-24-33941R4

Dear Dr. Etafa,

We’re pleased to inform you that your manuscript has been judged scientifically suitable for publication and will be formally accepted for publication once it meets all outstanding technical requirements.

Kind regards,

Chalachew Adugna Wubneh, MSc

Academic Editor

PLOS ONE

Additional Editor Comments (optional):

Dear authors,

Please address lines 90, 306-309, 324, and 361 during your proofreading.

Reviewers' comments:

Reviewer's Responses to Questions

**Comments to the Author**

Reviewer #2: All comments have been addressed

2. Is the manuscript technically sound, and do the data support the conclusions?

Reviewer #2: Yes

3. Has the statistical analysis been performed appropriately and rigorously?

Reviewer #2: Yes

4. Have the authors made all data underlying the findings in their manuscript fully available?

Reviewer #2: Yes

5. Is the manuscript presented in an intelligible fashion and written in standard English?

Reviewer #2: Yes

Reviewer #2: (No Response)

**Do you want your identity to be public for this peer review?** For information about this choice, including consent withdrawal, please see our Privacy Policy

Reviewer #2: No

---

## [Editor Report · Acceptance letter]

PONE-D-24-33941R4

PLOS ONE

Dear Dr. Etafa,

I'm pleased to inform you that your manuscript has been deemed suitable for publication in PLOS ONE. Congratulations! Your manuscript is now being handed over to our production team.

Kind regards,

on behalf of

Dr. Chalachew Adugna Wubneh

Academic Editor

PLOS ONE